# Mathematical Model Insights into EEG Origin under Transcranial Direct Current Stimulation (tDCS) in the Context of Psychosis

**DOI:** 10.3390/jcm11071845

**Published:** 2022-03-26

**Authors:** Joséphine Riedinger, Axel Hutt

**Affiliations:** 1MLMS, MIMESIS, Université de Strasbourg, CNRS, lnria, ICube, 67000 Strasbourg, France; josephine.riedinger@inria.fr; 2INSERM, U1114, Neuropsychologie Cognitive et Physiopathologie de la Schizophrénie, 67085 Strasbourg, France

**Keywords:** tDCS, ketamine, psychotic transition, EEG, modelling, thalamocortical circuit

## Abstract

Schizophrenia is a psychotic disease that develops progressively over years with a transition from prodromal to psychotic state associated with a disruption in brain activity. Transcranial Direct Current Stimulation (tDCS), known to alleviate pharmaco-resistant symptoms in patients suffering from schizophrenia, promises to prevent such a psychotic transition. To understand better how tDCS affects brain activity, we propose a neural cortico-thalamo-cortical (CTC) circuit model involving the Ascending Reticular Arousal System (ARAS) that permits to describe major impact features of tDCS, such as excitability for short-duration stimulation and electroencephalography (EEG) power modulation for long-duration stimulation. To this end, the mathematical model relates stimulus duration and Long-Term Plasticity (LTP) effect, in addition to describing the temporal LTP decay after stimulus offset. This new relation promises to optimize future stimulation protocols. Moreover, we reproduce successfully EEG-power modulation under tDCS in a ketamine-induced psychosis model and confirm the N-methyl-d-aspartate (NMDA) receptor hypofunction hypothesis in the etiopathophysiology of schizophrenia. The model description points to an important role of the ARAS and the δ-rhythm synchronicity in CTC circuit in early-stage psychosis.

## 1. Introduction

Schizophrenia is a psychotic disease that affects almost 1% of the worldwide population. This disorder has a multifactorial etiology which includes environmental and genetic factors, and develops progressively along years before to manifest generally between age 15 and 25. During the prodromal pre-psychotic phase, first, low-intensity psychotic/positive symptoms insidiously appear [1,2]. Examples of psychotic and negative symptoms are thoughts with paranoid tendencies or auditory verbal hallucinations and decrease of the psychomotor activity, respectively. About one-third of the at-risk mental state patients will experience the psychotic transition, marked by the intensification and worsening of psychotic symptoms over almost 2 years. This frequently culminates in an acute episode of psychosis, and finally results in the chronicisation of the psychotic syndromes [1,3,4]. For patients diagnosed with schizophrenia, pharmacological and psychological therapy allow to alleviate symptoms and significantly improve patients’ everyday life. Nevertheless, finding treatments to prevent the psychotic transition would be a precious lever for patients and psychiatry.

One of the promising noninvasive preventive approaches is transcranial Direct Current Stimulation (tDCS). It modulates neuronal activity by generating a constant current between two electrodes, and current polarity distinguishes anodal and cathodal tDCS (a-tDCS and c-tDCS) [5,6]. Electrophysiological studies showed that short-time tDCS applications, i.e., from seconds to a few minutes, act directly on the membrane resting potential of cortical neurons. Triggered excitability shifts and altered resting potentials are transient and depend of stimulation polarity. Hence, short a-tDCS induces an increased excitability and resting potential, whereas c-tDCS diminishes both excitability and resting potential, without long-term after-effects in both cases [5,7,8]. Conversely, stimulations of longer duration, i.e., several minutes and more, can elicit prolonged after-effects of several hours [6,9]. These plasticity effects require neural action evolving on a larger time scale than the synaptic short-time action, implicating complex dynamics of diverse ion channels, cascade signalling and protein transcriptions. Hence, while influence of c-tDCS is unclear and appears to result from mixed mechanisms of Long-Term Potentiation (LTP) and Long-Term Depression (LTD), a-tDCS long-term after-effects are probably mediated by LTP mechanisms [10,11,12].

The stimulation-triggered mechanisms are still poorly understood today. They elicit the potential of tDCS to modulate neuronal activity and in particular to treat impaired activities, as in neurological and psychiatric disorders. In consequence, numerous clinical studies of tDCS protocols for the treatment of various disease (i.e., schizophrenia, major depressive disorder and stroke) have been conducted, with controversial findings [13,14]. For instance, in patients suffering from schizophrenia, a robust reduction of negative symptoms and/or cognitive symptoms and/or auditory verbal hallucinations with low side-effects has been observed after application of a tDCS protocol of a few days [15,16,17] and, when a follow-up after a month was performed, effects were long-lasting (several months). In these studies, anodal electrodes are placed unilaterally or bilaterally on the dorsolateral prefrontal cortex. Nevertheless, other studies with similar stimulation setup and anodal electrode positioning have not found any significant improvement in symptoms of patients [18,19]. Despite a lack of consistency in the literature and a need to replicate studies, the use of tDCS could be efficient in the reduction of schizophrenic symptoms, mainly for auditory verbal hallucinations [13]. Additionally, the questions of whether and how psychosis can be prevented by tDCS remains open. Understanding better the mechanism of action and electrophysiological influence of tDCS on neuronal networks is now essential for the future design of optimal treatment and preventive stimulation protocols.

The dysconnection hypothesis in schizophrenia postulates an elementary etiological role for the alteration of functional synaptic connectivity, i.e., not an anatomical dysconnection but an aberrant modulation of synaptic efficiency. It allows to reveal links between the pathophysiology of the disease and clinical signs and indicates inter- and intrastructural impairments in functional integration processes [20,21]. Indeed, cortico-thalamo-cortical (CTC) dysfunctions and disturbance in functionally associated neuronal oscillations were reported from the pre-psychotic phase of schizophrenia. The CTC network is known to be highly implicated in attentional, sensorimotor and cognitive processes. These oscillopathies encompass an increased power of prefrontal resting state activity in the broadband γ frequency (30–80 Hz) [22,23], as well as a reduced power in cortical low-frequency oscillations, such as in the δ band (1–4 Hz) [24,25] and in non-REM sleep spindles occurring in the σ-frequency band (10–17 Hz) [26,27].

Interestingly, some of these prodrome-related oscillopaties can be transiently reproduced in humans and rodents by a sub-anaesthetic administration of the N-methyl-d-aspartate (NMDA) receptor antagonist ketamine. A single sub-anaesthetic dose of ketamine induces psychotomimetic effects in healthy humans [28] and worsens positive symptoms in patients suffering from schizophrenia [29]. Moreover, electrophysiological studies in rats [30,31,32] and humans [33,34,35] have shown that such a psychotomimetic ketamine administration disrupts the CTC functional dynamics, and induces an increased power of γ-oscillations and a reduced power of slow wave, an effect reversed by the antipsychotic action of clozapine [36]. Therefore, the sub-anaesthetic ketamine provides a reliable animal model of a transition to a psychosis-relevant state [37], and supports the dysconnection hypothesis for the etiopathophysiology of schizophrenia, more precisely, via a hypofunction of the NMDA receptors [38].

To describe the ketamine action in a mathematical model, it is essential to implement its physiological action. The anaesthetic ketamine induces several diverse neural actions [39,40,41]. For instance, it antagonizes NMDA-receptors in cortical and subcortical structure, increases the cortical and subcortical glutamate concentration and affects the cortical dopamine system in mammals. As a major effect, at sub-anaesthetic doses, it is known to enhance the excitability level in cortical and subcortical structures, possibly through disinhibition of excitatory cortical pyramidal neurons [42,43]. Since the balance between excitation and inhibition impacts the oscillatory brain activity, it is expected that ketamine modulates the EEG power spectrum. Indeed, ketamine induces increased gamma power and a decreased power in lower delta frequency bands in humans [33] and rats [30,31,36,43]. Moreover, a general δ-power reduction has been observed in the entire CTC circuit [36,43]. Recent experimental studies on a slow-wave sleep model in rats have shown a decrease of frontoparietal spectral power in the σ-frequency band [36,44]. The present work aims to reproduce these latter ketamine-induced modifications in brain rhythms.

Although a large number of experimental studies have revealed multiple aspects of tDCS action in humans and animals, it is still not clear how the external electric currents affect neural information processing. To improve tDCS further and develop new stimulation protocols, advanced knowledge and understanding on the underlying neural dynamics is advantageous. One approach aims to extract brain models from experimental data, such as finite-element tissue models [45,46,47]. These models permit to extract the spatial localisation of tDCS-induced electric flow in the tissue, but they do not describe the neural rhythmic activity which represents an essential marker in mental diseases. Other studies have described such neural rhythms under tDCS. For instance, the authors of [48] focus on cerebellum tDCS and show how tDCS affects cerebellum firing. Moreover, [49] considers the full brain, taking into account a realistic connectivity matrix, but neglects certain frequency ranges and is unspecific to the pre-clinical condition.

We intend to describe neural rhythms in the major frequency range 1–60 Hz and relate them to experimental data under pre-clinical conditions. To this end, we present a mathematical population model that both considers biological properties of the neural population and reproduces qualitatively experimental findings. Such a mathematical model permits to test diverse hypotheses on neural action and validates some of them. In this context, our model approach is based on the insight that tDCS has been shown to improve the health situation of a large number of subjects of different ages, nationalities and environments. Hence, the underlying mechanism of tDCS response has to be universal in some sense and not subject-specific. We implement a rather unspecific neural model that only considers, e.g., the essential brain topography of mammals and the known impact of drugs on synaptic receptors. A successful model reveals the essential elements that play a major role in the tDCS response. More specifically, we ask why a long-term stimulation impact of tDCS vanishes in the course of days and weeks, whereas it takes a much shorter stimulation time to induce a long-term impact. In this context, it is not clear why and how different inter-stimulus intervals, or more generally diverse stimulation protocols, determine the impact duration of tDCS. The present work proposes a plasticity model that explains the impact of inter-stimulus on the impact duration. Moreover, the diminution of γ activity by tDCS in a ketamine animal model of psychotic transition [44] is unclear. As it is desirable to be able to diminish the pathological γ−activity in human pre-psychotic patients, at first, it is important to understand why tDCS can diminish γ activity in a ketamine model. This may provide deeper insights into ways of improving tDCS.

In the present work, we hypothesize that a rather subject-unspecific but biologically reasonable mathematical model is sufficient to describe the tDCS impact on mammalian brain. To this end, we propose to explain the impact of anodal tDCS on psychotic-like brain dynamics by the use of a mathematical model of the CTC circuit. This model encompasses the CTC connectivity topology, as well as excitatory and inhibitory neuronal populations in the network structures. At first, we reproduce typical activity modulations by tDCS, such as excitability and EEG power spectrum, to validate the neural model. Then, the psychosis network dynamic is mimicked by modelling the action of ketamine on glutamatergic neurotransmission. Finally, tDCS application in the context of psychosis is simulated and forecasts of tDCS action on CTC structures are provided. The neuromodulatory action of tDCS is modelled by a cortical constant input and a change in the neuronal population transfer function and EEG is computed numerically for all conditions and compared with literature. We shed light on the essential neural mechanisms and CTC interactions that allow to mimic prodrome-related oscillopathies and the observed effect of tDCS in psychotic patients.

## 2. Materials and Methods

### 2.1. The Model

It is well established that EEG originates from population activity in cortical layers [50,51]. Neural ensemble models are well known to provide a very good description of EEG [51,52,53,54]. The employed model describes population activity in the supragranular cortex layers I–III (SG) and the cortico-thalamic loop between granular/infragranular cortical layers IV–VI (GIG), the thalamic relay cells and the reticular structure, see Section A.1 for details. The network is sketched in Figure 1. SG cells receive input from the brainstem, or more generally, from the Ascending Reticular Arousal System (ARAS) [55,56], and cells in input layer IV exhibit afferent connections from thalamic structures and connect to infragranular cortical cells in layers V and VI (Figure 1A). Moreover, the GIG cells receive input from thalamic relay cells in the CTC and project back to thalamic relay and reticular cells, cf. Figure 1B. GABAergic neurons in the reticular nucleus receive excitatory inputs from GIG pyramidal cells and thalamic relay neurons, while the inhibitory reticular outputs are sent to the thalamic relay cells.

The principal brain rhythms investigated in the present study are the δ (1–4 Hz), σ (10–17 Hz) and the γ rhythm (30–80 Hz). Following previous modelling studies [53,57,58,59], we hypothesize that the δ and σ rhythms originate in the cortico-thalamic circuit (including both thalamic and GIG structures), whereas γ rhythms are generated in the SG layers [59,60] in accordance to experimental findings [61]. The proposed model considers interactions of excitatory and inhibitory local networks as the major rhythm origin, in line with previous modelling studies [62,63].

### 2.2. The Action of Transcranial Direct Current Stimulation

As previously stated, to understand and describe the action of tDCS by a model, it is important to take into account tDCS polarity as well as duration of the current administration. The present work implements the action of short tDCS, i.e., from seconds to a few minutes, over neuronal excitability and membrane resting potentials by a transient steepness shift of transfer functions of both neuron types in all cortical layers. Simultaneously, a constant input in these cortical layers is applied. As demonstrated in previous studies [60,64], the synchronous impact on both input and transfer function steepness is a direct result of a modulation by Poisson spiking activity in the ensemble, cf. the Section A.2. Hence, simulating its influence over neuronal excitability and resting potential, a-tDCS (c-tDCS) is implemented by an increase (decrease) of constant input and steepness decrease (increase) of transfer functions for both excitatory and inhibitory cortical neurons. In order to show the excitability effect by tDCS, evoked potentials are simulated by a transient external stimulation. System response magnitude and its baseline level subject to the model tDCS current are then evaluated. Here, we mimic a-tDCS and c-tDCS currents by model currents $I_{\mathrm{tDCS}}>0$ and $I_{\mathrm{tDCS}}<0$, respectively.

The after-effects of long stimulation, i.e., several minutes or longer, are primarily implemented for a-tDCS. The overall complex prolonged plasticity effect triggered by a-tDCS is described by an enhanced excitatory synaptic efficacy, mimicking the LTP impact. To this end, we introduce a factor $f_{\mathrm{tDCS}}$ for the excitatory synaptic efficacy in both supra- and infra-granular layers, cf. the Section A.2 for more details.

### 2.3. The Action of Ketamine

We implemented the ketamine modulation of excitation and inhibition on neural populations to reproduce the EEG power spectrum changes observed in previous experimental studies. Specifically, we consider the ketamine-induced NMDA hypofunction at GABAergic neurons in the cortex and thalamus. This represents an effective disinhibition of glutamatergic neurons, cf. the Section A.4.

### 2.4. Functional Connectivity

Our model network includes connected neural populations. To learn more on the interaction among these populations, we evaluate the functional connectivity (FC) between them. FC quantifies the time-dependent degree of interaction between two populations. Several measures have been proposed to quantify it [65], such as spike-field coherence [58] or other synchronization indices. We consider here the time-averaged phase coherence [58,66] which represents the phase-locking value (PLV) [67] between two time series with 0≤PLV≤1. These time series represent the activity of different populations in different frequency bands. It has been shown recently [68] that phase synchronization quantifies the degree of information sharing in noise-driven models. For PLV=1, two time series are completely phase-locked and their populations activity is maximum synchronized, whereas PLV=0 reflects a complete activity desynchronization. The larger the phase coherence PLV, the larger the functional connectivity between two populations and the more the two populations interact.

## 3. Results

### 3.1. Excitability

Excitability modification is one of the major features of short-term tDCS. Figure 2 shows model simulations of an event-related potential (ERP) (A) and its impact on the resting state activity (B). We observe an augmented excitability for a-tDCS in the ERP. For both GIG cortical and thalamic relay neurons, short-term tDCS induce an increased (decreased) resting membrane potential for a-tDCS (c-tDCS). Conversely, the resting firing rate is poorly affected.

### 3.2. Anodal tDCS-Effect on Spectral Power

Long duration a-tDCS stimulation is known to yield cognitive enhancement [69], induces LTP [10] and may serve an important role in clinical psychiatry in the future [15,16,17,70]. To describe the involved plasticity, we have developed a model that relates tDCS duration and plasticity effect. Figure 3A (left) shows the major model idea: for short stimulation duration, no plasticity is present, while the impact increases with longer stimulation and saturates after some time. After a-tDCS offset, the plasticity reduces but much more slowly, see Figure 3A (right). The understanding of both enhanced and diminished plasticity in the course of time is important since a-tDCS stimulation protocols alternate between stimulation with a certain duration and current intensity, and stimulation pauses. Our temporal model postulates the tDCS impact on plasticity strength subjected to stimulation time and stimulation pauses. Figure 3B shows the impact of the single stimulation duration in a stimulation sequence on the plasticity, similar to the tDCS studies described in [6]. We observe an increase in plasticity strength $f_{\mathrm{plast}}$ with increasing stimulation duration $T_{s}$ and hence a prolonged long-term effect after stimulation offset. For details on the numerical implementation, please see the Section A.2.

The plasticity effect is accompanied by spectral power enhancement, as shown in Figure 4. We demonstrate that long-duration a-tDCS induces EEG power enhancement in the δ-, σ- and γ-frequency range, which decreases with time after the stimulation ends.

The model simulations also permit to predict population activity in the thalamic structures, cf. Figure 5. We observe a relay cell population activity, which is very similar to the EEG activity. Conversely, the reticular population exhibits a much weaker γ activity but a strong δ activity. The a-tDCS appears to have no impact on the activity of thalamic structures.

### 3.3. Impact of Anodal tDCS in Psychosis Model

Previous experimental studies on humans [33,34] and rats [36,72,73] under ketamine have shown that the experimental setup represents a reasonable psychosis model. The subjects exhibit pathological ketamine-induced enhanced γ and diminished σ and δ activity [30,31,32,33,34,44]. These spectral features resemble the power modulation in at-risk mental state patients for psychosis. It has also been shown experimentally that an additional long-duration anodal tDCS stimulation reverses the impact of ketamine [44] which resembles the clinical impact of a-tDCS in human psychosis patients. Figure 6 presents EEG and subcortical activity in the control condition (no ketamine, no a-tDCS), under ketamine but no a-tDCS, and in the presence of ketamine and a-tDCS. Under ketamine, the EEG power decreases in the δ- and σ-frequency range but increases in the γ-frequency range. The relay cell population responds similarly but the ketamine impact is less prominent. Conversely, in the reticular cell population, ketamine enhances slightly spectral power in the δ-frequency range only.

Now, applying a-tDCS enhances EEG power in the δ- and σ-frequency band and moves the spectral EEG power from the γ-frequency band to lower frequencies. Relay cells respond to the a-tDCS by a similar power increase in the δ- and σ-frequency, while γ-activity is retained. In the reticular population, the a-tDCS stimulation enhances the δ power but does not affect the spectral power in larger frequencies.

These ketamine-induced effects are summarized in Figure 7A, where cortical and relay cell power decrease in the δ- and σ-frequency band with ketamine, while ketamine enhances δ power in reticular cells. Additional a-tDCS stimulation enhances further cortical and relay cell population power in the δ and σ band, while the impact on reticular population is weak. Conversely, γ activity is strongly enhanced by ketamine in cortical cells and diminished by a-tDCS. The sub-cortical γ activity is poorly affected by the a-tDCS application.

To reveal more details on the brain dynamics in psychosis, functional connectivity between brain areas provide some insights [74,75]. We have computed the phase coherence in the CTC between their involved neural populations, cf. Figure 7B. Ketamine enhances FC in the δ-frequency range between the cortical GIG, the relay and the reticular cells, while it diminishes FC in the σ range between these populations. The FC is also reduced between cortical and thalamic relay cells and between the thalamic structures in the γ-frequency range under ketamine, whereas it does not affect the cortico-reticular connectivity. Now adding a-tDCS, FC increases between almost all areas and frequency bands, except in the intra-thalamic connections in the γ band. Hence, a-tDCS inverses the ketamine FC reduction in the σ band between all areas and in the γ range between GIG layer cells and relay cells.

## 4. Discussion

### 4.1. Excitability and Spectral Power

We present a neural population model which permits to reproduce major experimental EEG data features observed under tDCS. This model implies a cortico-thalamic feedback loop that generates the δ and σ rhythm. This loop includes GIG cortical cells, relay cells and reticular cells with corresponding excitatory and inhibitory synaptic connections. Moreover, a SG cell population is the origin of the γ rhythm in good accordance with previous experimental results [76,77]. The reproduced features include an increased (decreased) excitability for a-tDCS (c-tDCS) in good agreement with previous experimental findings [5,7,8]. The presented excitability study is very brief and demonstrates a fundamental excitability modulation with tDCS. Future work should evaluate the presented evoked potential model much further, e.g., comparing modelling results to previous detailed experimental results [78].

In addition, long-duration a-tDCS enhances EEG power in all frequency bands of interest, which is in good agreement with experimental studies in healthy humans [79,80,81,82,83]. Our model assumes LTP-like long-term potentiation of excitatory synapses and corresponding model simulations reproduce well the EEG power enhanced by a-tDCS. We propose to describe the rather complex neurophysiological plasticity effects by a simple modulation of the synaptic excitatory efficacy. Moreover, we present a stimulation-timing model that relates a temporal stimulation protocol with the plasticity effect. Future studies will evaluate this model by comparison with experimental tDCS effects, as described in [84,85].

Since this model simplification reproduces EEG power features successfully, the present work supports the LTP-like effect of a-tDCS. The model also permits to predict the a-tDCS impact on thalamic cells and we find a poor effect on the thalamic power. Future experimental work will evaluate and verify this finding. As an additional well-known effect, long-term impact of tDCS on the brain decays after some time. This effect is central in patients whose possibly successful tDCS impact decay after some months after treatment stop [6,15,16,17]. We have proposed a decay model of the plasticity effect which describes well the modulation of the tDCS after stimulation ends, cf. Figure 3 and Figure 4.

Several previous studies have modelled the electric field induced by tDCS [45,46,86]. Such studies provide some insights into the spatial location of tDCS impact, but do not allow to explain the dynamic impact on experimentally observed data. A different recent model study has focused on the tDCS impact on single neuron activity in neural populations considering homeostatic structural plasticity [87]. This impressive work takes into account different stimulation protocols in more detail than our work does. However, it does not explain major tDCS effects on neural oscillatory brain activity that have been described at length in previous studies. The present work provides such an explanation in the context of psychosis, taking into account recent neurophysiological hypothesis. Moreover, the ensemble model does not exhibit a realistic brain topology such as our CTC model and hence is less realistic concerning the brain network structure.

### 4.2. Spectral Power in the Psychosis Model

We have further extended the neural model to describe EEG under psychotic transition assuming a ketamine model [35,37]. Identifying the brain state induced by ketamine with the psychotic state and the control brain state as a healthy state, we observe a drop of δ and σ power in the simulated EEG in the psychotic-like state compared to the healthy state. Conversely, the psychotic-like state exhibits a strong γ power enhancement. These findings are in good agreement with experimental data in humans [33,34,35] and rats [30,31,32]. Nevertheless, we also show an increase in the δ power of reticular cells in the simulated psychotic state. This result differs from recent findings in which all CTC structures present a δ-power decrease after acute ketamine treatment [36,43]. Additionally, previous experimental studies have shown that a-tDCS may alleviate symptoms in psychosis patients [15,16,17,88]. For pre-psychotic patients, our model predicts an enhancement of δ and σ power and a reduction of γ power by a-tDCS, which well replicates previous experimental findings in an animal ketamine model [44].

### 4.3. Connectivity Modulation in the Psychosis Model

The functional connectivity between brain areas reflects the brain’s ability to pass and share information between the structures. For instance, brain networks in mental disorders exhibit diminished FC between brain areas and, thus, a network fragmentation [89]. Enhanced functional dysconnectivity has been found in early psychosis [90] and reduced conscious access in psychosis is related to diminished long-range structural connectivity [91], thus supporting the dysconnection hypothesis in schizophrenia [20,21]. This fragmentation can be reversed by psychotherapy in combination with a placebo or anti-psychotics [92] and by a-tDCS [75]. It is interesting to note that a-tDCS not only enhances FC in psychosis patients but also in healthy subjects [93]. This indicates that a-tDCS may enhance brain FC in general. We have investigated this effect by computing FC in our neural model and found equivalent results. In the σ and γ frequency range, FC decreases in the ketamine model compared to the control condition. Conversely, FC between all CTC structures increases in the δ frequency band during the ketamine condition suggesting a δ hypersynchronisation in the entire CTC network in the psychotic state. This new result adds to the previous finding of intra-thalamic γ-synchronisation [36] driven by cortical “γ-noise” [32]. Interestingly, for all frequency bands, a subsequent a-tDCS stimulation re-enhances FC. The only exception is the intra-thalamic FC in the γ range that decreases by a-tDCS.

### 4.4. Strength and Limits of the Neural Model

The proposed model permits to reproduce experimental results on excitability and spectral power of EEG originating in cortical layers, as well as excitability and spectral power of neural activity in non-cortical brain areas. Moreover, the model allows to predict FC in the cortico-thalamic circuit and elucidates how tDCS augments information sharing between brain areas. To gain these insights, the neural model considers recent hypotheses on how tDCS affects synapses and neuronal populations. For instance, we assume that tDCS induces an intrinsic Poisson-noise activity in single neurons that represents a modulated input current and transfer function in the population [64]. This allows to describe the well-known modulated excitability by short-duration tDCS. Moreover, the simple assumption that long-duration anodal tDCS reflects an augmented excitatory synaptic efficacy in cortical neurons similar to the effect of LTP permits to describe the EEG power modulation under a-tDCS.

To describe the anodal tDCS-impact in psychosis-related prodrome, we present a neural ketamine model that considers both the hypofunction of NMDA receptors and the ketamine-induced disinhibition. Both mechanisms appear to be essential to reproduce the σ-power diminution and γ enhancement found experimentally. Since ketamine both inhibits excitatory cells and inhibitory cells, and the latter inhibition yields disinhibition of the population, it is an open question how the different power responses emerge. Our model explains this by actions in two sub-circuits. Neuronal populations in the cortico-thalamic loop, including granular/infragranular cortical layer, thalamic relay cells and reticular neurons, experience primarily the ketamine-induced hypofunction generating the power-diminished σ-rhythm. Simultaneously, the cortical supragranular layer neurons and the ARAS are affected by the ketamine-induced disinhibition. Since the thalamus and ARAS are known to project to the supragranular layer neurons generating the γ-rhythm [59], the ketamine-induced disinhibition yields a γ-power enhancement, as observed experimentally.

Recalling that a-tDCS has been found to enhance the brain, it appears counterintuitive that the additional a-tDCS diminishes the γ power in the ketamine model, as shown experimentally [44]. To resolve this seeming contradiction, our simulations indicate that the combination of the cortical Poisson-neuron input and ARAS input in the supragranular layers yields a power shift from the γ-frequency range to lower frequencies, and hence a reduction in γ power. This power shift explains the γ-power drop, but remains to be shown experimentally.

To the best of our knowledge, the proposed model fills a gap in the current literature on tDCS models. It bridges physiological action of tDCS and ketamine on the microscopic scale and the tDCS-population response (EEG) on the macroscopic scale. Additionally, it reproduces EEG power spectra in a broad range of frequencies, targets the psychotic disorder as pre-clinical condition and captures information processing aspects such as the enhanced functional connection under tDCS linking to behaviour. Previous model studies have focussed on the spatial distribution of electric currents [45,46,47], but have not distinguished neural rhythms as the present work. Such rhythms have been investigated, e.g., in the context of cerebellum tDCS [48], focussing on the firing rate change in this area, but not on a broad frequency band as in the present work. In [49], the authors consider the full brain, taking into account a realistic connectivity matrix, but focus on lower frequencies only and do not consider pre-clinical conditions. Moreover, the response of neural populations on tDCS was investigated in [94], where the authors focused on evoked potentials and not on spectral power distributions and did not consider a specific mental disorder.

The proposed model considers the cortico-thalamic feedback loop and implements the neurophysiological action of ketamine. A recent exciting work [95] also reproduces EEG features in a broad range of frequencies under ketamine by employing a cortico-thalamic population model. The authors found indications that ketamine primarily affects the connections between SG and GIG cells. This insight is interesting, while the present work assumes ketamine impact both in the cortical SG cells and the thalamus. Moreover, the present work also considers tDCS action in contrast to [95].

A different work [87] assumes that long-term stimulation makes the stimulated population grow in size, which yields enhanced activity. The present work describes the corresponding power enhancement by long-term potentiation. The population growth hypothesis does not explain the frequency-specific impact of tDCS under ketamine, whereas the proposed model hypothesizes that the frequency-specific enhancement under ketamine is controlled by the ARAS input in the SG layers.

The presented model is limited to a rate-coding population model that considers firing rates of single neurons but neglects the timing of single spikes and spike train modes. This simplification is reasonable in the description of mesoscopic population activity such as Local Field Potentials or EEG [54]. However, tDCS has been shown to modulate both spike patterns and spike timing [96] and may affect the spectral power of neural populations without modulating neuron firing rates [97]. Our neural model does not describe such a spike train effect, making it unable to capture such rhythm modulations. This represents a limit of the population rate model.

Moreover, the model does not specify the underlying neuron type but implies a general type-I neuron [98]. It also does not specify details of intra- and inter-network connections, such as realistic spatial neural networks in single brain areas and topographic thalamo-cortical mapping. At first glance, this may represent a strong limitation of the model and questions its validity. However, we argue that we are interested in the general underlying neural mechanism of tDCS and its impact in psychosis valid for unspecific mammalian subjects. Since tDCS in different subjects of different mammalian species trigger similar EEG neural responses [6,11,70,71], it is important to choose a general rather simplistic population model that captures essential neural features and hence is independent of specific model choices. Here, the essential features are the nonlinear transfer function between synaptic input and population firing rate, realistic synaptic response models and a brain network topology of realistic polarity, i.e., with realistic excitation and inhibition. For instance, the emergence of a reticular δ-activity observed in Figure 4 matches experimental findings in the thalamus [99,100] and results from the chosen network topology, as shown previously [53].

### 4.5. Perspectives

Future work will elaborate further on the plasticity model for long-term tDCS effects presented in Figure 3. To this end, previous experimental studies on the impact of stimulation protocol parameters [84,85] will provide the data to fit parameters in the plasticity model. This fit will yield an optimal hands-on plasticity model that permits to control better the long-term impact of tDCS. Moreover, future experimental studies of the γ and β spectral (17–30 Hz) power under tDCS will reveal whether reduced γ activity implies enhanced β activity, as seen in Figure 6. Such studies will validate or falsify the underlying hypothesis that the ARAS contributes to the γ reduction.

The model also provides the possibility to implement an adaptive closed-loop feedback control framework [101,102]. This adaptive real-time control will determine an optimal stimulation, e.g., to diminish γ activity in pre-psychotic patients.

Future work should also improve the neural model by including neural mechanisms of spindles, since these are known to represent important EEG markers in psychosis and their prodrome [36,103,104]. Incorporating spiking modes of the reticular population should allow to reproduce such dynamics, switching from bursting mode in the control/healthy condition to single-action potential/tonic mode in the psychosis/ketamine condition [36]. In addition, there is strong evidence that the δ rhythm originates in the cortico-thalamic loop [105,106] and its role in psychosis remains to be revealed [107]. Future work will address the function and origin of δ rhythms in psychosis in more detail. Essentially, the present work demonstrates how to describe mathematically neurostimulation impact on EEG and how to predict brain activity. This may trigger models for other neurostimulation techniques over the scalp, such as transcranial Alternating Current Stimulation (tACS), transcranial Magnetic Stimulation (TMS) or cerebellar neurostimulation [48]. Indeed, most neuromodulatory approaches have attracted increasing attention in recent years and seem to represent promising tools for future clinical treatment and prevention in psychiatry [108].

## 5. Conclusions

The present work provides a mathematical description of tDCS impact on brain activity in the context of psychosis and aids in improving experimental setups. To reproduce the ketamine-induced oscillopathies, usually used to mimic the schizophrenic prodrome-related EEG, the essential neural mechanisms are a reduced cortical inhibitory efficacy and an increased ARAS input to cortical structures. This supports the NMDA receptor hypofunction hypothesis in schizophrenia. Moreover, the simulation of tDCS influence on EEG can be achieved by simple modulation of synaptic efficacy, with a temporal dimension to mimic long-term plasticity effects. Finally, these findings support the idea that a-tDCS could reduce, or even normalize, schizophrenia prodrome-related oscillopathies, indicating its powerful potential as a preventive treatment.

## Figures and Tables

**Figure 1 jcm-11-01845-f001:**
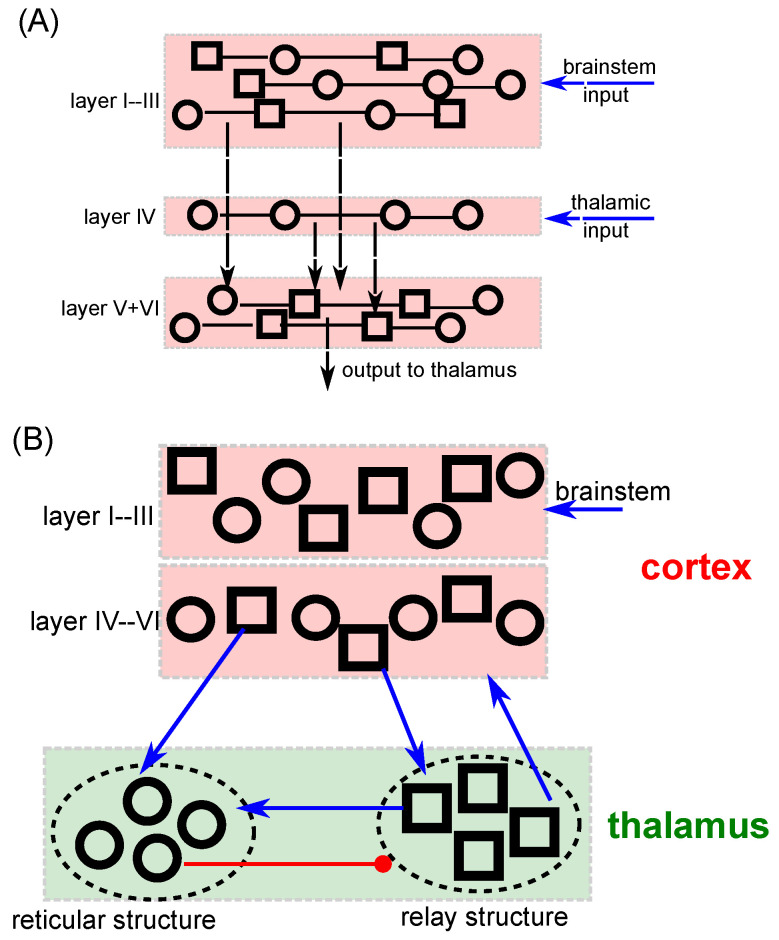
Cortico-thalamo-cortical network model of neural populations. (**A**) The supragranular layers I–III, granular layer IV and infragranular layers V and VI exhibit excitatory (represented by squares) and inhibitory (circles) neurons. (**B**) Blue and red connections between cortex and thalamic structures denote excitatory and inhibitory synaptic connections. The reticular and relay structures includes inhibitory (circles) and excitatory (squares) neurons, respectively.

**Figure 2 jcm-11-01845-f002:**
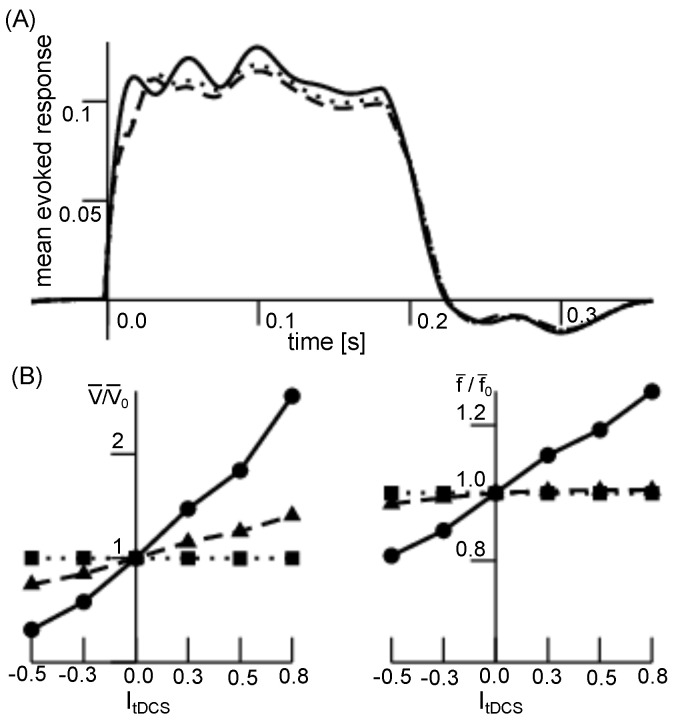
tDCS-induced excitability. (**A**) The evoked potential time course is shown for tDCS input current $I_{\mathrm{tDCS}}=0.0$ (dashed), $I_{\mathrm{tDCS}}=0.3$ (dotted) and $I_{\mathrm{tDCS}}=0.8$ (solid). The larger $I_{\mathrm{tDCS}}$, the larger is the response magnitude during transient stimulation (up to ∼200 ms). After stimulation, the system response is almost identical for all input currents. (**B**) The relative baseline resting potential (left panel) and firing rate (right panel) in the precursor time interval 50 ms before stimulation input. The value ${\bar{V}}_{0}\left({\bar{f}}_{0}\right)$ is the baseline resting potential (resting firing rate) for absent input currents. The lines denote the summed up relative resting activity in the cortical GIG neurons (solid line), in the excitatory relay cells (dashed line) and the reticular cells (dotted line). We observe an increase of the resting membrane potential (left) for $I_{\mathrm{tDCS}}>0$ (a-tDCS) and a decrease for $I_{\mathrm{tDCS}}<0$ (c-tDCS) in cortical and relay cells, whereas no impact on reticular cells is found. Conversely, the resting firing rate is poorly modulated by the tDCS current.

**Figure 3 jcm-11-01845-f003:**
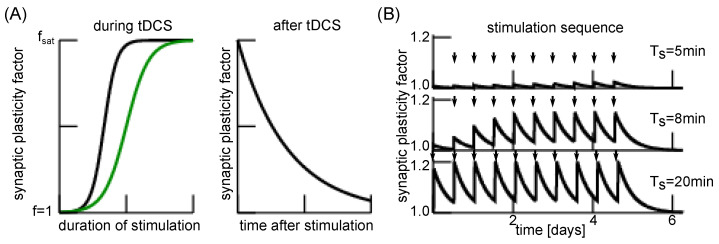
Modelled plasticity impact during and after anodal tDCS stimulation. (**A**) Left panel: very short stimulus duration does not induce any plasticity (the synaptic plasticity factor is f≈1), medium duration exhibits a strong plasticity effect ($1\leq f\leq f_{\mathrm{sat}}$) that is not enhanced anymore for long-duration stimuli [6] ($f\to f_{\mathrm{sat}}$) with saturation factor $f_{\mathrm{sat}}$. For high and low a-tDCS currents (black and green curve, respectively), the qualitative behaviour is similar, but it takes more time to induce the same plasticity effect for lower a-tDCS currents [71]. Our model describes this temporal evolution of plasticity effect by a population growth model, see Section A.2 for more details. Right panel: after stimulation offset the plasticity effect diminishes exponentially with time. Experimental studies [71] indicate that the decay time scale is in the range of tens of minutes for typical previous stimulation current (∼0.5–1 mA). (**B**) Simulation of the plasticity effect in a typical a-tDCS sequence motivated by [6]. Arrows indicate stimulation periods. A single a-tDCS period had a duration $T_{s}$ with a stimulation pause of 12 h, 10 repetitions and a final period of 34 h, cf. Section A.2 for more details. Further parameters are $\tau_{\mathrm{plast}}=3$ h and $\tau_{\mathrm{decay}}=500$ h.

**Figure 4 jcm-11-01845-f004:**
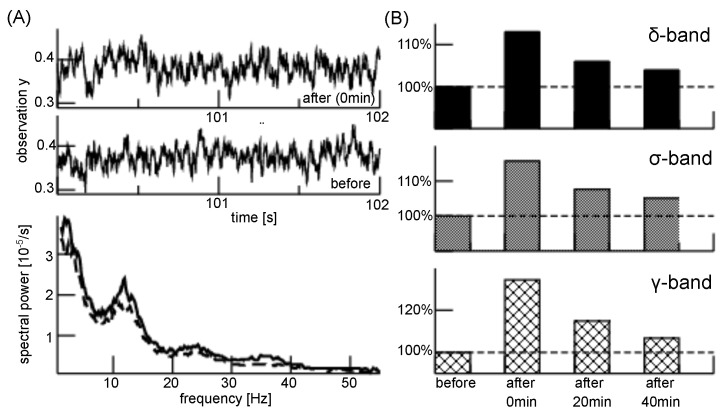
Long-duration anodal tDCS impact on EEG. (**A**) Upper panels: simulated EEG time traces before a-tDCS stimulation and immediately after. Lower panel: the corresponding power spectral density distributions for activity before (dashed line) and immediately after (solid line) stimulation. (**B**) Relative power change with respect to power before stimulus for different time periods after stimulation. Plasticity effects are modeled by an increase of excitatory efficacy with factors $f_{\mathrm{tDCS}}=1.2$ (0 min after stimulation), $f_{\mathrm{tDCS}}=1.1$ (20 min after stimulation) and $f_{\mathrm{tDCS}}=1.05$ (40 min after stimulation). We assume a stimulation duration time of 12 min, implying the plasticity time scale $\tau_{\mathrm{plast}}=1$ min and a decay of the plasticity effect with $\tau_{\mathrm{decay}}=30$ min. In addition, the excitatory synaptic efficacy to ARAS input is $f_{\mathrm{tDCS}}^{\mathrm{resp}}=f_{\mathrm{tDCS}}$.

**Figure 5 jcm-11-01845-f005:**
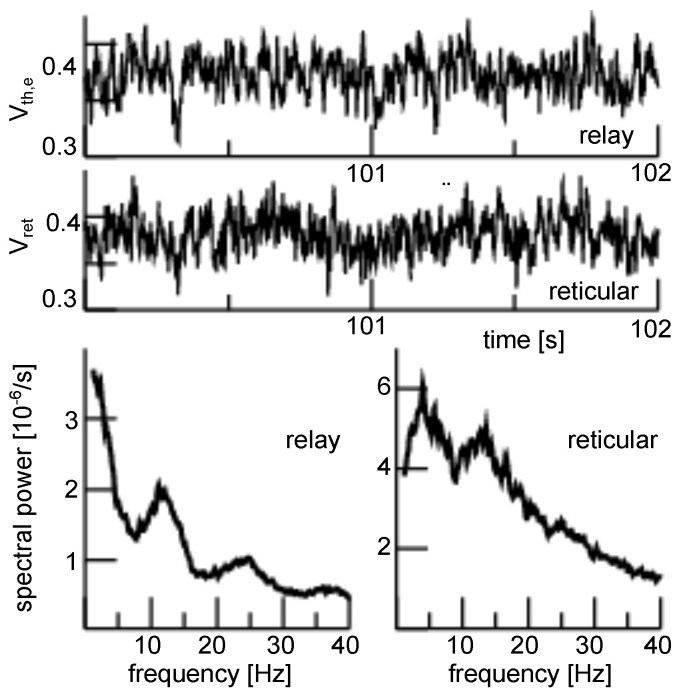
Long-duration anodal tDCS impact on subcortical activity. Upper panels: time series of simulated population activity for a-tDCS stimulation with $f_{\mathrm{tDCS}}=1.2$. Lower panels: spectral power density distribution of relay and reticular population activity in the absence (dashed line, $f_{\mathrm{tDCS}}=1.0$) and presence (solid line, $f_{\mathrm{tDCS}}=1.2$) of a-tDCS stimulation.

**Figure 6 jcm-11-01845-f006:**
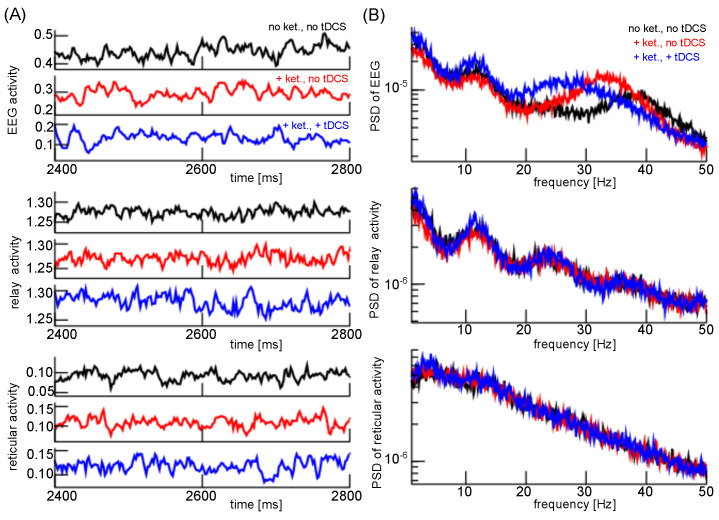
Cortical and subcortical activity under ketamine and anodal tDCS. (**A**) Model activity time courses of EEG (upper panels), relay cell population (center panels) and reticular cell population (lower panels) under different conditions. (**B**) Power spectral density distributions of EEG (top panel), relay cell population panel (center panel) and reticular cell population (bottom panel). Parameters are $f_{\mathrm{tDcs}}=1.05$ corresponding to stimulation duration of 4 min with $\tau_{\mathrm{plast}}=1$ min, $f_{\mathrm{ketamine}}=0.7$ in the cortico-thalamic loop and $f_{\mathrm{ketamine}}=0.8$, $f_{\mathrm{tDcs}}^{\mathrm{resp}}=2.0$ in SG layers.

**Figure 7 jcm-11-01845-f007:**
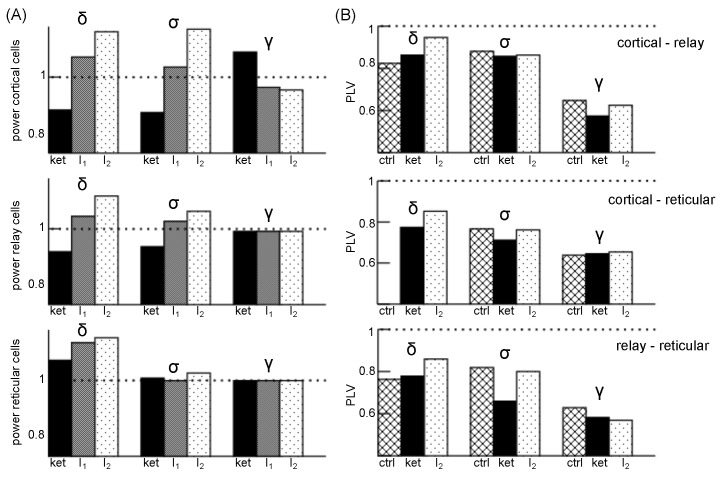
Relative spectral power and phase coherence PLV in the cortico-thalamic circuit. (**A**) The ratio of power spectral density averaged over frequencies in the corresponding frequency band and the corresponding average power spectral density in the control condition. The power in cortical cells consider GIG cells. (**B**) Degree of phase coherence (PLV) between excitatory cortical GIG population activity and the excitatory relay population activity (top panel), the excitatory cortical GIG layer and the reticular population activity (center panel) and between the reticular and relay cells (bottom panel). Note that there is no cortico-reticular δ-phase coherence in the control condition. Model parameters are $f_{\mathrm{tDCS}}^{\mathrm{resp}}=2.0$, $f_{\mathrm{tDCS}}=1.03$ and $f_{\mathrm{tDCS}}=1.05$ for stimulation input $I_{1}$ and $I_{2}>I_{1}$, respectively. Ketamine parameters are identical to parameters used in Figure 6.

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
