# Peer review of "Mathematical Model Insights into EEG Origin under Transcranial Direct Current Stimulation (tDCS) in the Context of Psychosis"

_jcm, 2022, doi:10.3390/jcm11071845_

Round 1

Reviewer 1 Report

Title: I suggest adding the term 'mathematical' to the title to avoid any confusion.

Introduction:

Riedinger and Hutt propose a mathematical model of neuronal circuits to simulate EEG power in the delta, alpha and gamma band when submitted to tDCS following ketamine treatment.

In general, the introduction reads well, but the third paragraph could be improved as it currently overestimates the level of knowledge owned by the reader. For example, please briefly describe the "dysconnection hypothesis in schizophrenia".

Line 53, typo to 'osccillopaties'

Methods:

First paragraph of section 2.2 should be integrated to the introduction.

P4, line 114 refers to Poisson spiking activity, directing the reader to the appendix, but there is no further explanation there. As it is now, this paragraph is difficult to read as many concepts are introduced, but not adequately explained.

When referring to the appendix, please specify which section of the appendix is referred to.

Similar to 2.2, first paragraph of 2.3, as well as part of the second paragraph, belongs to the introduction. I understand that it may be convenient to present some background information when stating your methodology, but it is also disruptive. As a reader, when I look at the methods section, I want to know what you've done, not why you've done it. I'll look at the introduction for the 'why'. For 2.3, lines 136 to 147 could be summarised in one sentence.

Please specify which software was used to run your models.

Results:

Here again, I find elements of literature. This section should only present result. Please keep the comparison with literature to the discussion. This will allow the reader to clearly identify your own results.

Discussion:

"The proposed model permits to reproduce experimental results on excitability and spectral power of EEG originating in cortical layers, as well as excitability and spectral power of neural activity in non-cortical brain areas." This is great. What I'm missing in the discussion is 'what's next?'. What can we use this model for. This should also make its way to the abstract.

Section 4.4 should point towards the strengths and limitations in a more straightforward way. For example, what is the strength or limitation presented in the third paragraph?

Conclusion:

"these findings support the idea that a-tDCS could reduce, or even normalize, schizophrenia prodrome-related oscillopathies, indicating its powerful potential as a preventive treatment." Considering that this is a mathematical model programmed to reproduce observations made in the living human or animal, I'm not certain one can draw this conclusion.

Reviewer 2 Report

In their manuscript entitled „Model insights into EEG origin under transcranial direct current stimulation (tDCS) in the context of psychosis", Joséphine Riedinger and Axel Hutt present a study in which a mathematical model is used to explore the effectiveness of the transcranial direct current stimulation to restore the Cortico-Thalamo-Cortical excitability dysfunctions that characterize the prodromal stages of schizophrenia.
In my opinion, this is an interesting and timely study. It seems well designed, well conducted and well written in most of its parts. I see just a series of minor issues to be fixed:

  • Authors at the end of the introduction section clearly summarize what they intend to explain and describe in the rest of the main manuscript. However, I think that they should detail this part by specifying objectives (main and secondary) and eventual hypothesis.

  • Linked to the previous point, I suggest Authors to specify the gap (s) of the literature that motivates the aims/hypothesis of their study.

  • Authors in the introduction section cite previous literature about the effectiveness of the transcranial direct current stimulation. However, as clearly described in recent guidelines and updates, it’s efficacy is not established for a significant number of pathological conditions and the data are still controversial. Therefore, I suggest Authors to discuss di point, citing papers like:

Fregni, F. et al. Evidence-based guidelines and secondary meta-analysis for the use of transcranial direct current stimulation (tDCS) in neurological and psychiatric disorders. Int. J. Neuropsychopharmacol. (2020).

Lefaucheur, J.-P. et al. Evidence-based guidelines on the therapeutic use of repetitive transcranial magnetic stimulation (rTMS): An update (2014–2018). Clin. Neurophysiol. (2020).

Lefaucheur, J.-P. et al. Evidence-based guidelines on the therapeutic use of transcranial direct current stimulation (tDCS). Clin. Neurophysiol. 128, 56–92 (2017).  

Lefaucheur, J.-P. et al. Evidence-based guidelines on the therapeutic use of repetitive transcranial magnetic stimulation (rTMS). Clin. Neurophysiol. 125, 2150–2206 (2014).

Reviewer 3 Report

There are grammatical errors and I suspect English is not their native language.

• The limitations of the proposed model should be highlighted

• How the method deals with real time applications, please discus

• Authors need to explain why they are using this methods. It is not clear (usefulness)

• Please explain the novelty of the proposed method compared with the state of the art.

• Discussion: Highlight the advantages and disadvantages of your method. Compare with other works.
